# Electrochemical Synthesis of Nb-Doped BaTiO_3_ Nanoparticles with Titanium-Niobium Alloy as Electrode

**DOI:** 10.3390/nano13020252

**Published:** 2023-01-06

**Authors:** Qi Yuan, Wencai Hu, Tao Wang, Sen Wang, Gaobin Liu, Xueyan Han, Feixiang Guo, Yongheng Fan

**Affiliations:** 1School of Materials and Metallurgy, University of Science and Technology Liaoning, Anshan 114051, China; 2Fujian Huaqing Electronic Material Technology Co., Ltd., Quanzhou 362000, China; 3Jiangsu Can Qin Technology Co., Ltd., Suzhou 215633, China

**Keywords:** Nb-BaTiO_3_, electrochemical synthesis, microstructure, nanoparticles

## Abstract

In this paper, Nb-doped BaTiO_3_ nanoparticles (BaNb_0.47_Ti_0.53_O_3_) were prepared using an electrochemical method in an alkaline solution, with titanium-niobium alloy as the electrode. The results indicated that under relatively mild conditions (normal temperature and pressure, V < 60 V, I < 5 A), cubic perovskite phase Nb-doped BaTiO_3_ nanoparticles with high crystallinity and uniform distribution can be synthesized. With this increase in alkalinity, the crystallinity of the sample increases, the crystal grain size decreases, and the particles become more equally dispersed. Furthermore, in our study, the average grain size of the nanoparticles was 5–20 nm, and the particles with good crystallinity were obtained at a concentration of 3 mol/L of NaOH. This provides a new idea and method for introducing foreign ions under high alkalinity conditions.

## 1. Introduction

BaTiO_3_ is a significant component of electronic functional ceramics. It is frequently utilized in electronic devices because of its excellent dielectric, piezoelectric and ferroelectric characteristics [1,2]. With the rapid development of electronic information and fifth-generation industry, electronic components are developing in the direction of high integration, multi-functioning, and miniaturization; higher requirements are put forward to prepare BaTiO_3_ nanomaterials that meet the performance requirements [3,4].

At present, the most common preparation methods of BaTiO_3_ materials mainly include the solid-state method [5] and a range of chemical synthesis techniques for preparing ultrafine BaTiO_3_ powders, such as the polymer precursor [6,7,8], hydrothermal, precipitation [9], sol-gel [10,11,12], and other methods. What is worthy of recognition is that chemical synthesis technology can fundamentally optimize the performance of electronic materials, especially in terms of the grain-size effect mechanism [13,14,15,16]. Despite these advantages, there are also many defects, such as long reaction times, demanding equipment requirements, and high costs. In contrast, the electrochemical synthesis reaction device is simple, and grain size control is achieved by varying the current density and the applied potential. Tao et al. [1] reported an electrochemical method in which BaTiO_3_ nanoparticles were synthesized in a H_2_O/EtOH (ethyl alcohol) solution containing KOH and Ba(OH)_2_, using two titanium plates as electrodes. The results indicated that the composition of the electrolyte affects the size of the BaTiO_3_ nanoparticles. Subsequently, Santos et al. [17] investigated the influence of solvent type and different electrolyte compositions on the synthesis of BaTiO_3_ nanoparticles. The results showed that highly crystalline nanoparticles could be synthesized using ethanol and methanol in a high alkalinity environment, while changing the alkalinity of the electrolyte might viably control the size of the synthesized nanoparticles.

In order to meet the performance requirements of materials in the industry and practical applications, doped BaTiO_3_-based ceramics are mostly used [18]. For instance, the anti-reduction performance of MLCC dielectric materials can be improved by doping Mn^2+^, Ca^2+^, and Mg^2+^ [19,20,21]. The modification of Curie points by introducing foreign ions into BaTiO_3_ ceramics is also an important means by which to study the preparation of widely temperature-stable MLCC. This group of common dopants includes La, Nb, Ce, and Zr ions [22,23,24,25,26,27,28]. The traditional solid-state method and a series of wet chemical synthesis methods to synthesize doped barium titanate powder have a long preparation cycle and time-consuming process, whereas the electrochemical method can provide a quick, simple, and one-time way to synthesize doped BaTiO_3_ powder.

So far as we know, the electrochemical synthesis of doped BaTiO_3_ nanopowders with alloy material as the electrode has not previously been reported. Hence, we report for the first time the electrochemical synthesis of Nb-doped BaTiO_3_, using titanium–niobium alloy plates as the electrode. The influence of the alkalinity of electrolyte on the grain size, phase composition, and morphology of the Nb-doped BaTiO_3_ is investigated. The results provide a novel method for the electrochemical synthesis of doped BaTiO_3_ powder.

## 2. Materials and Methods

The schematic diagram of the electrochemical synthesis of Nb-doped BaTiO_3_ nanoparticles is shown in Figure 1. The titanium–niobium alloy plate (purity > 99.0%, Xinbiao Metal Materials Co., Jiangsu, Wuxi, China) with a thickness of 1 mm was cut into a rectangle of 50 mm × 20 mm, to be used as an electrode for the electrolytic reaction. The surface of the electrode was mechanically polished with 800 Cw and 1200 Cw sandpaper. After cleaning with ultrapure water and EtOH, the electrode was washed with an ultrasonic wave in acetone to remove the remaining oil on the surface of the electrode. A heat-resistant glass electrolytic cell with a capacity of 250 mL is used for the electrolysis reaction. The electrolyte is composed of 50 mL EtOH, 150 mL ultrapure water (pre-boiled for 30 min to remove as much CO_2_ as possible), 0.05 mol·L^−1^ Ba(OH)_2_, and different concentrations of NaOH (0.5 mol·L^−1^, 1 mol·L^−1^, 1.5 mol·L^−1^, 2 mol·L^−1^, 2.5 mol·L^−1^, 3 mol·L^−1^) composition. The titanium-niobium alloy plate was fixed to the electrode holder, then two-thirds of its length was immersed in the electrolyte at a distance of 20 mm. The synthesis process uses a DC power supply (RXN-650D) and magnetic stirring. The initial voltage is 60 V, and the current is kept constant at 3.25 A.

The obtained powder sample was washed with 0.1 mol·L^−1^ of hot dilute nitric acid to remove the impurity BaCO_3_ and was then repeatedly filtered and washed with ultrapure water and ethanol. The sample was placed in an electrothermal constant temperature blast-drying box and dried at 100 °C for 2 h. Nano BaTiO_3_ (purity 99.9%, size < 100 nm, Aladdin, Shanghai, China) was selected as the reference material.

The crystal phase of the synthesized powder was analyzed by X-ray diffraction (Panalytical X’Pert-MPD type, Cu Kα radiation source, 40 kV, 40 mA). All scans were carried out at 5°/min for 2θ values between 10° and 90°. Small-angle X-ray scattering (SAXS) was used to calculate the grain size, based on the X-ray scattering intensity from the sample, measured at a scattering angle from 0.1° to 5°. The chemical composition of the samples was analyzed using a monochromatic Al Kα X-ray source and X-ray photoelectron spectroscopy (XPS, Thermo Scientific K-alpha, MA, USA). All XPS spectra were calibrated in terms of C1s at 284.8 eV. An ultra-high-resolution field emission scanning electron microscope (SEM, Zeiss, Ober-Kochen, Germany), energy spectrometer (EDS), and spherical aberration-corrected transmission electron microscope (TEM, HF5000, 200 kV) were used to observe the morphology, grain size, and microstructure of the powder.

## 3. Results and Discussions

Figure 2 shows the XRD diffraction pattern of the prepared BaNb_0.47_Ti_0.53_O_3_ (hereafter referred to as BNTO) powder and pure BaTiO_3_. The obtained powders all have a cubic perovskite ABO_3_ structure (a = b = c, α = γ = β = 90°) and are consistent with the inorganic crystal structure database ICSD standard card (PDF-98-008-3902). Figure 2b shows a partially enlarged view of the XRD patterns of the two samples in the range of 30°~50°. The diffraction peak position of the BNTO powder has changed to a low angle compared to that of BaTiO_3_. The phenomenon can be explained by the Bragg equation:(1)2dsinθ=nλ
where d is the interplanar spacing, n is an integer, called the reflection order, and θ is the grazing angle (the ionic radii of Nb^4+^ and Ti^4+^ are 0.74 Å and 0.61 Å, respectively [29,30]). The Nb^4+^ ion radius is larger; the interplanar spacing likely becomes larger after replacing part of Ti^4+^ at the B site, which causes the diffraction peaks to shift to a low-angle direction (Figure 2b).

To prove the above content, XPS measurement was performed on the synthesized BNTO nanoparticles. Figure 3a shows the XPS spectra of Ti2p, Nb3d, Ba3d, and O1s, indicating that the sample contains barium, titanium, and niobium. Figure 3b,c shows the fine spectra of Ti2p and Nb3d energy levels, respectively. It can be seen from the spectrum that the binding energies of the Ti element in Ti2p_1/2_ and Ti2p_3/2_ are 460.4 eV and 454.7 eV, which correspond to the two characteristic peaks of Ti^4+^ [31]. The binding energies of 3d_3/2_ and 3d_5/2_ are the 205.8 eV and 203.1 eV Nb3d doublet, respectively, as observed in the fine spectrum of the Nb3d energy level in Figure 3c. After querying the standard reference database, NIST, we found two characteristic peaks corresponding to Nb^4+^ [32] that were consistent with the conclusions drawn regarding the reason for the shift of XRD diffraction peaks mentioned above. The Nb/Ti ratio of 4.84/5.36 was obtained, using Avantage to fit the fine XPS spectra of Nb and Ti. XPS calculation is a semi-quantitative method, but the Nb/Ti ratio is very close to BaNb_0.47_Ti_0.53_O_3_.

The relationship between the breakdown potential, the size of the crystal grain of the synthesized powder, and the NaOH concentration (mol·L^−1^) during the reaction process of electrochemically synthesized BNTO powder is shown in Figure 4 and Table 1. The crystallite size of the sample was obtained via the Guiner curve-fitting of SAXS data (See Appendix A for SAXS data map). When the NaOH concentration increases from 1 mol·L^−1^ to 3 mol·L^−1^, the breakdown potential gradually decreases; the increase in the concentration of NaOH in the electrolyte means that the conductive anions and cations increase, and the current conduction becomes easier, meaning that the electrochemical synthesis reaction can be completed with a lower applied potential. From Figure 4 and Table 1, it can be observed that the grain size and breakdown potential have the same trend concerning the change in NaOH concentration. During the reaction, the appearance of anode sparks is very important for forming BNTO nanoparticles [1]. As NaOH concentration in the electrolyte increases, the reaction becomes gentler, and anode sparks can be more evenly distributed on the electrode surface, meaning that the Ti and Nb are better separated and are distributed from the immersion area of the electrode.

Figure 5 shows the time-varying curves of the potential and current under different NaOH concentrations during the synthesis of BNTO nanoparticles. When the NaOH concentration is 1 mol·L^−1^, the required applied potential is higher, and the potential decreases significantly with the increase in concentration. Under the conditions of a single concentration, the potential was initially high. However, the potential decreased with the extension of the reaction time, and it was more stable after the reaction had proceeded for 40 min. Since the total surface area of the alloy, the electrode is large and the reaction is violent at the beginning of the reaction, the required potential is higher. After a while, the alloy electrode gradually dissolves, which leads first to a decrease in total surface area and then the decline of potential. This phenomenon can indicate that the high alkalinity environment of the electrolyte can ensure the current conduction during the whole reaction process, thereby promoting the effective separation of Ti and Nb from the electrode (as shown in Figure 6). The anodic reaction can be described as follows (2) and (3) [33,34]: (2)$$0.47\mathrm{Ti}+0{.53\mathrm{Nb}+6\mathrm{OH}}^{-}\overset{\mathrm{spark}}{\to }{{[\mathrm{Nb}}_{0.47}{\mathrm{Ti}}_{0.53}O_{3}]}^{2-}{+3H}_{2}O+4e\prime$$
(3)$${\mathrm{Ba}}^{2+}{+[\mathrm{Nb}}_{0.47}{\mathrm{Ti}}_{0.53}O_{3}{]}^{2-}\overset{}{\to }{\mathrm{BaNb}}_{0.47}{\mathrm{Ti}}_{0.53}O_{3}\left(s\right).$$

With the increase in NaOH concentration, the grain size of the BNTO nanoparticles gradually decreases, which may be due to the more stable current conduction, milder reaction, and more uniform distribution of the anode spark on the electrode in the environment of a higher concentration of NaOH. The change makes the better separation and distribution of Ti and Nb at the interface between the electrode and electrolyte, promoting grain-size reduction.

Figure 7 shows the XRD patterns of samples prepared with different concentrations of NaOH (1/2/3 mol·L^−1^). It can be seen that when the concentration of NaOH is low (1 mol·L^−1^), obvious diffraction peaks of BaCO_3_ will appear at 2θ = 24.27° and 34.06°. The intensity of the diffraction peak gradually weakens or even disappears. This means that the high-alkalinity environment will inhibit the generation of BaCO_3_ impurities. The ultrapure water in the electrolyte has been boiled before being used, to remove as much CO_2_ as possible. Although CO_2_ in the air may still be dissolved in the electrolyte during the reaction (the solubility of CO_2_ in water at room temperature and pressure at 0.033 mol·L^−1^), the temperature of the electrolyte will increase and the solubility of CO_2_ will decrease during the reaction. In addition, a small amount of CO_3_^2−^ will preferentially neutralize with OH^−^ in an alkaline environment, thereby inhibiting the formation of BaCO_3_. The lattice parameter, cell volume, and average crystallite size were obtained by fitting the XRD data (a full spectrum fitting using the JADE (MDI. Jade. 6.0) software). The calculation results are shown in Table 2. With the increase in NaOH concentration, the lattice constant of the sample increases and the crystallite size decreases; the crystallite size is very close to the SAXS calculation result.

Figure 8a–c shows the SEM images of BNTO powder when prepared under different NaOH concentrations. It can be observed that a small number of amorphous particles appear when the NaOH concentration is low, indicating that the nucleation and crystal growth are insufficient and immature under these conditions, resulting in the obvious heterogeneity and certain agglomeration of the sample. With the increase in NaOH concentration, the particle shape becomes clearer. When the NaOH concentration is at 3.0 mol·L^−1^, the particle shape is the most complete. Figure 8d and Table 3 show the EDS spectrum and quantitative composition of point A in Figure 8c. In Table 3, L/K represents the line system of the characteristic X-rays. The actual atomic ratio of Ti:Nb is almost equal to the stoichiometric ratio of 1:1.13 in BaNb_0.47_Ti_0.53_O_3_, which indicates that Nb-doped BaTiO_3_ powder was synthesized via the electrochemical method, using the titanium-niobium alloy material as the electrode. EDS data show that Ba:Ti:Nb = 1:0.69:0.96, and the proportion of Nb and Ti elements is higher than expected. By observing Figure 8d, it is clear that Ba and Ti peaks appear as “pathological overlaps” near the positions of 4.5 keV and 4.8 keV. This kind of “pathological overlap” may cause identification errors when detected by the EDS equipment, which makes the proportion of the Ti element higher than expected and the proportion of the Ba element lower than expected, resulting in an inconsistent proportion of elements when analyzed by EDS.

To further study the morphology and microstructure of the BNTO powder, TEM observation was carried out. According to Figure 9a (TEM image), the particle size of the powder was roughly estimated to be about 5–10 nm, which was in good agreement with the particle size as measured by small-angle X-ray scattering (SAXS). It was observed that the powder has a certain degree of agglomeration, which may be due to the small grain size, high surface energy, and surface tension, which causes agglomeration and the formation of larger aggregates [35]. Figure 9b (TEM local magnification image) shows the measured values of lattice fringes at 0.291 and 0.239 nm, which correspond to the spacing of the (110) and (111) crystal planes of BNTO, respectively. The XRD diffraction peaks (110) and (111) (2θ angles of 28°~33° and 37°~40°) were fitted and calculated by applying origin software; the results were consistent with the interfacial distance values in the HRTEM images (NaOH concentration at 3.0 mol·L^−1^). The clear lattice fringes in the TEM image also illustrate the synthesis of the BNTO nanoparticles, which have high crystallinity.

## 4. Conclusions

Nb-doped BaTiO_3_ nanoparticles were synthesized successfully by the electrochemical method using titanium-niobium alloy as the electrode, which overcomes the disadvantage that most metal ions are difficult to dissolve under high alkalinity and inhibit the electrochemical synthesis reaction. Compared with other Nb-doped BaTiO_3_ synthesis methods, the experimental device of this scheme is simple, and well-crystallized BaNb_0.47_Ti_0.53_O_3_ nanoparticles can be obtained in a short time under a normal temperature and pressure. Among them, the concentration of NaOH has an important effect on the potential change during the entire synthesis reaction, and the crystallinity and grain size of the sample. When the concentration of NaOH increases from 1 mol·L^−1^ to 3 mol·L^−1^, the average grain size of the sample changes between 20 nm and 5 nm. Finally, the electrochemical synthesis of Nb-doped BaTiO_3_ nanoparticles using alloy materials as electrodes is expected to provide a simple method for the synthesis of other metal element-doped titanate nanoparticles.

## Figures and Tables

**Figure 1 nanomaterials-13-00252-f001:**
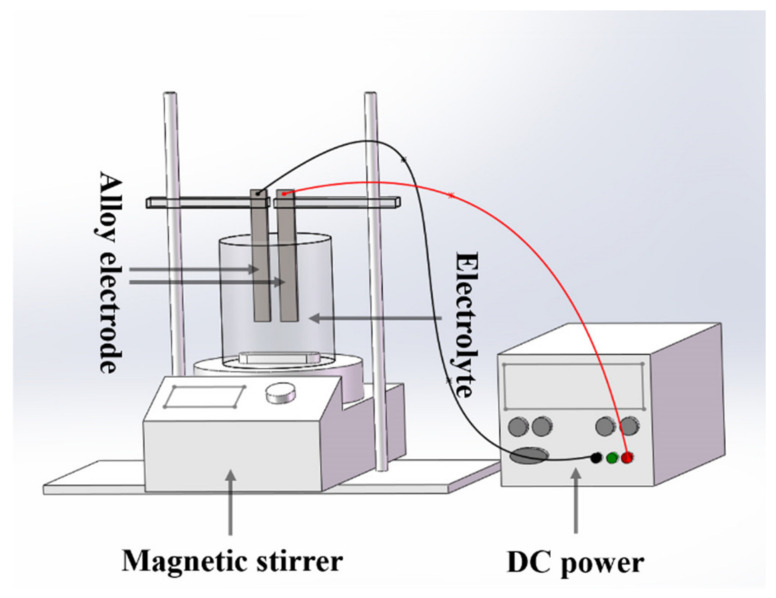
Schematic diagram of the electrochemical electrolytic cell.

**Figure 2 nanomaterials-13-00252-f002:**
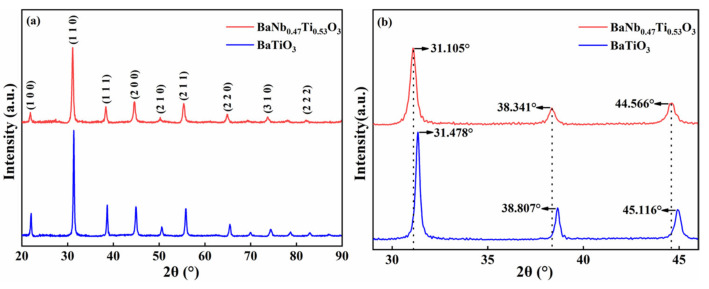
XRD pattern of electrochemically synthesized powder (**a**) and its partially enlarged schematic diagram (**b**).

**Figure 3 nanomaterials-13-00252-f003:**
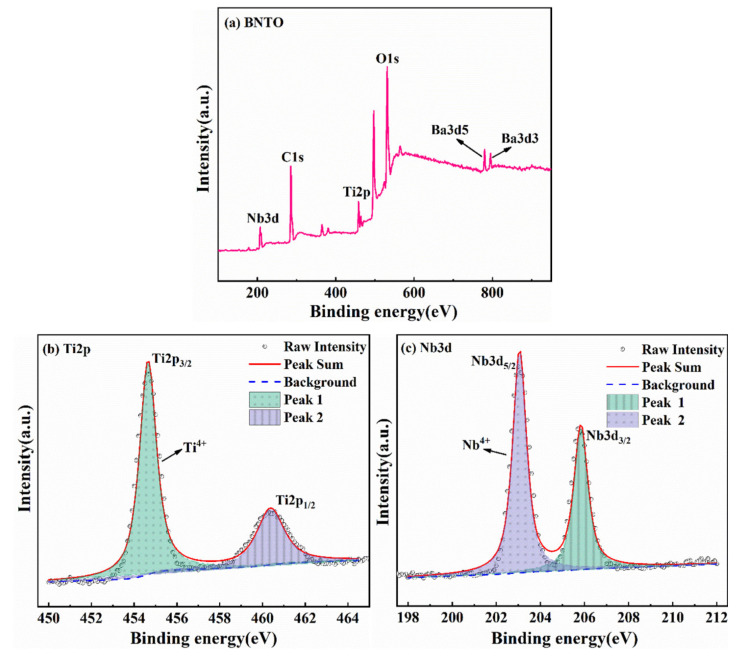
XPS spectra of electrochemically synthesized BNTO nanoparticles: (**a**) the XPS full spectrum of the sample, (**b**) the Ti2p fine spectrum, and (**c**) the Nb3d fine spectrum.

**Figure 4 nanomaterials-13-00252-f004:**
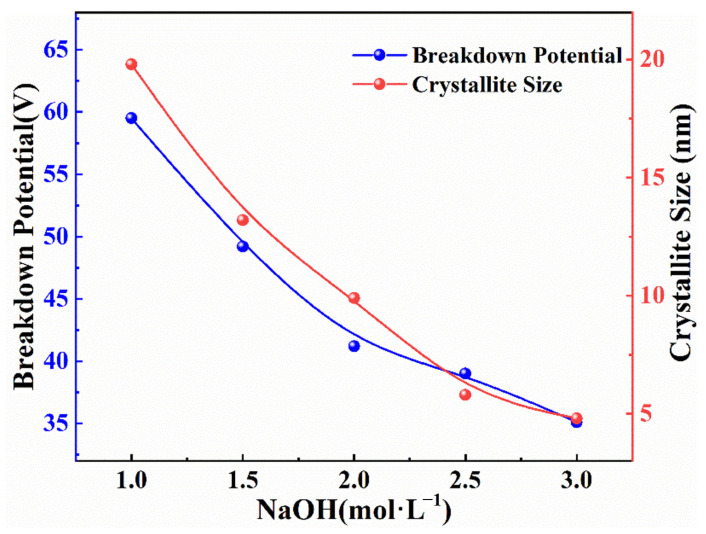
Relationship between breakdown potential, grain size, and NaOH concentration.

**Figure 5 nanomaterials-13-00252-f005:**
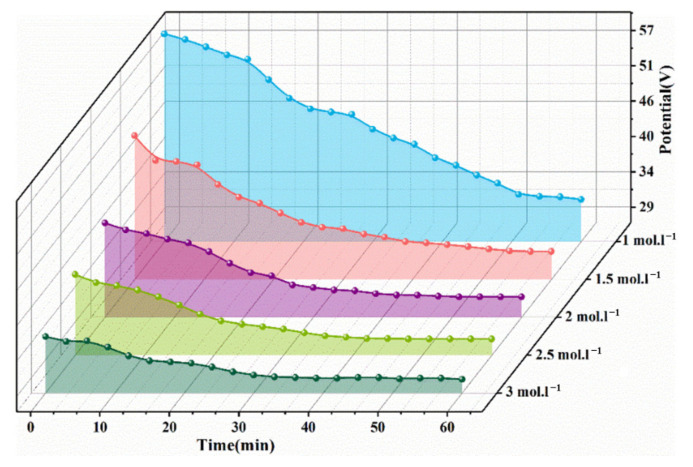
Relationship curve between potential and reaction time in the process of synthesizing BNTO nanoparticles under different NaOH concentrations.

**Figure 6 nanomaterials-13-00252-f006:**
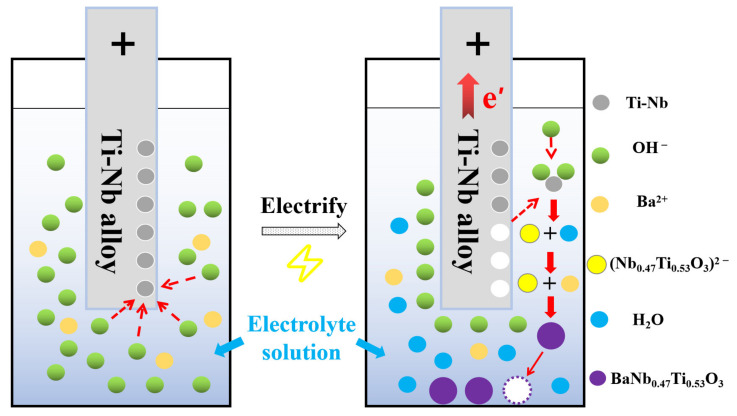
Anodic chemical reaction diagram.

**Figure 7 nanomaterials-13-00252-f007:**
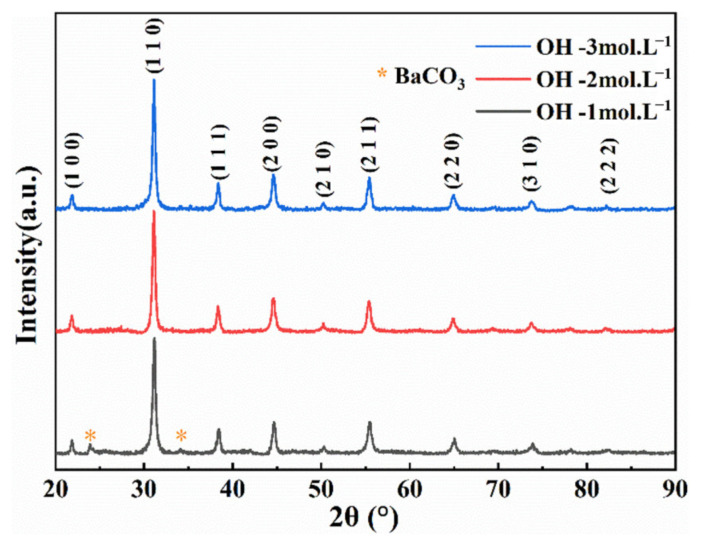
XRD patterns of samples prepared under different NaOH concentrations.

**Figure 8 nanomaterials-13-00252-f008:**
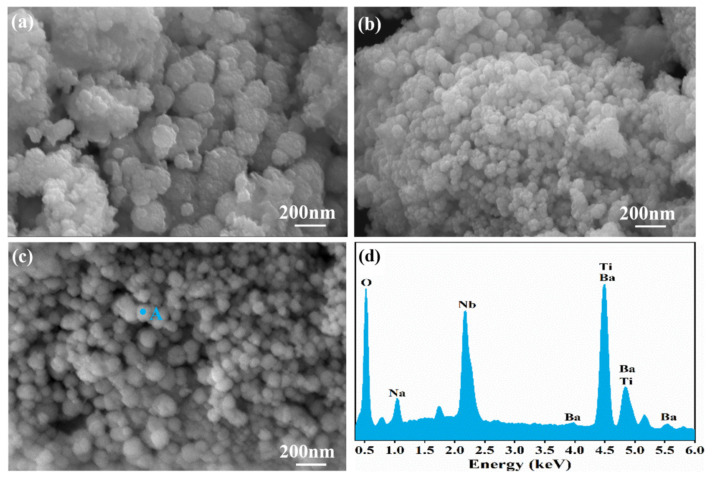
SEM image of BNTO powder, synthesized in solvents containing different concentrations of NaOH: (**a**) 1.0 mol·L^−1^ (**b**) 2.0 mol·L^−1^ (**c**) 3.0 mol·L^−1^; (**d**) EDS spectrum in point A.

**Figure 9 nanomaterials-13-00252-f009:**
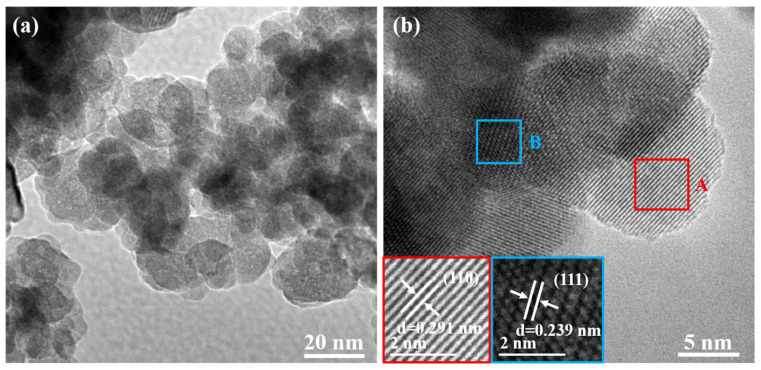
TEM image (**a**), local magnification image (**b**), crystal fringe image A and B of the electrochemically synthesized BNTO nanoparticles (NaOH concentration is 3.0 mol·L^−1^).

**Table 1 nanomaterials-13-00252-t001:** Effects of different NaOH concentrations on breakdown potential and the grain size of the synthetic powder.

Breakdown Potential (V)	NaOH (mol·L^−1^)	Initial Current (I)	Crystallite Size (nm)
59.5	1	3.25	19.8
49.2	1.5	3.25	13.2
41.2	2	3.25	9.9
39.0	2.5	3.25	5.8
35.1	3	3.25	4.8

**Table 2 nanomaterials-13-00252-t002:** The lattice parameter, cell volume, and crystallite size of the samples, prepared at different NaOH concentrations.

NaOH (mol·L^−1^)	a (Å)	V = a^3^ (Å^3^)	Crystallite Size (nm)
1	4.043 ± 0.010	66.1 ± 0.5	20.3 ± 1.6
2	4.059 ± 0.005	66.9 ± 0.2	10.5 ± 1.5
3	4.063 ± 0.006	67.1 ± 0.3	5.4 ± 1.0

**Table 3 nanomaterials-13-00252-t003:** The EDS spectrum analysis results of point A in Figure 8c.

Elements	Weight (%)	Atomic (%)
O K	30.11	71.04
Na K	1.96	3.05
Ti K	12.59	9.37
Nb L	17.6	6.75
Ba L	37.74	9.79
Totals	100	100

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
