# Peer review of "Electrochemical Synthesis of Nb-Doped BaTiO3 Nanoparticles with Titanium-Niobium Alloy as Electrode"

_nanomaterials, 2023, doi:10.3390/nano13020252_

Round 1

Reviewer 1 Report

The manuscript describes synthesis of the Ba(Nb,Ti)O3 nanoparticles by means of electrochemical oxidation of the Nb-Ti alloy. Authors performed systematic investigation of the synthetic conditions and found those that allow performing this synthesis in a way that is economically viable and yields high quality nanoparticles.

While main conclusions of the research look correct, description of the experiment is incomplete and should be elaborated more before the article can be published. More specifically,

1. Authors claim that the particle size was determined by means of SAXS. However, SAXS data are not shown anywhere in the text and the way they were treated is not described.

2. Nb/Ti ratio was determined by XPS. Again, nothing is said about the way how that was done, obtained values and standard deviations. Furthermore, atomic percentages of the metals obtained from EDS data significantly differ from those expected from the composition given by the authors (Ba:Ti:Nb=1:0.69:0.96 instead of 1:0.47:0.53). This difference should be explained.

3. Analysis of the XRD data is purely qualitative. At least lattice parameter of the obtained phase must be calculated. Apart from that, quality of the XRD data looks good enough to perform independent estimation both of crystallite size and Ti/Nb ratio, so I would highly recommend to perform these calculations.

Apart from the main problems listed above, some proofreading of the manuscript is necessary to improve its readability. Several problems are listed below.

4. Both tables are captioned as 'Table 1'

5. Line 44: "high crystallinity BaTiO3 nanoparticle" - should be "highly crystalline"

6. Line 47 tells "there are few reports on the synthesis of doped BaTiO3 powder by the electrochemical method" and line 58 tells "electrochemical synthesis of doped BaTiO3 nanopowders with alloy material as electrode has not been reported". These two phrases are contradictory.

7. Line 75-77: "Fix the titanium-niobium alloy plate to the electrode holder and submerge two-thirds of its length in the electrolyte at a 20 mm distance." This sentence should be rewritten from imperative to declarative.

8. Line 85-86: "Nano BaTiO3 was selected as the experimental control group." One substance cannot be regarded as a group. "Reference material" is the correct phrase.

Based on these observations, I recommend major revision of the manuscript before it can be accepted for publication.

Reviewer 2 Report

Authors propose a novel experimental protocol for obtaining Nb-doped BaTiO3 nanoparticles using an electrochemical method. Authors tested different experimental conditions and selected the most optimal ones. The presented scheme is simple and allows obtaining nanoparticles of different sizes in a short time under normal temperature and pressure. The article is well-written, figures and tables are illustrative, the logical flow of the article is clear. Overall, the article can be accepted for publication.

Minor: Table 2 is named Table 1; in Table 2, column 1 – the L/K designations should be explained.

Author Response

Point 1:Table 2 is named table 1; In Table 2, column 1 - the L/K designations should be explained.

Response: We apologize for our oversight. The manuscript table headings have been revised and the full-text drawings and tables examined closely. And explain L/K (Lines 204 to 205 of the manuscript).

Round 2

Reviewer 1 Report

The quality of the manuscript significantly improved after authors' revision. However, some points mentioned in the first review remain to be addressed properly.

1. SAXS data are still not shown. If authors consider amount of figures too high, I would recommend moving all experimental data to Supplementary Materials section and keep only results in the main text.

2. There is no sign of metallic Nb or Ti neither at XRD nor at TEM. So some other explanation is necessary for inconsistent element proportions obtained by EDS.

3. Calculation of structure parameters from XRD is not described properly. How many peaks were used for fitting? Were instrumental parameters (zero shift, sample displacement, instrumental peak broadening etc.) taken into account? Definitely, peak shape cannot be pure Gaussian for nanosized particles.

So additional corrections are necessary for the article to be acceptable for publication.

Author Response

Please find the reply in the attachment.

Round 3

Reviewer 1 Report

Quality of the data analysis significantly improved in the new version of the manuscript and now the data obtained by different techniques look consistent. The article can be published.

One minor recommendation is that it makes no sense to provide so many digits in Table 2 given the values of standard deviations. Data should be presented with no more than 1 significant digit of standard deviation unless this digit is equal to 1, where 2 significant digits are acceptable (that is, V=66.1±0.5, not 66.1059±0.5065 or a=4.043±0.010, not 4.0434±0.0103).

Author Response

Thank you for your suggestion. We have adjusted the data in Table 2 of the manuscript.